# Effectiveness of Protein Supplementation Combined with Resistance Training on Muscle Strength and Physical Performance in Elderly: A Systematic Review and Meta-Analysis

**DOI:** 10.3390/nu12092607

**Published:** 2020-08-27

**Authors:** Noé Labata-Lezaun, Luis Llurda-Almuzara, Carlos López-de-Celis, Jacobo Rodríguez-Sanz, Vanessa González-Rueda, César Hidalgo-García, Borja Muniz-Pardos, Albert Pérez-Bellmunt

**Affiliations:** 1Department of Basic Sciences, Faculty of Medicine and Health Sciences, Universitat Internacional de Catalunya, 08195 Sant Cugat del Vallès, Barcelona, Spain; nlabata@uic.es (N.L.-L.); lllurda@uic.es (L.L.-A.); jrodriguezs@uic.es (J.R.-S.); 2Physiotherapy Department, Faculty of Medicine and Health Sciences, Universitat Internacional de Catalunya, 08195 Sant Cugat del Vallès, Barcelona, Spain; carlesldc@uic.es (C.L.-d.-C.); vgonzalez@uic.es (V.G.-R.); 3Physiotherapy Department, Faculty of Health Sciences, Universidad de Zaragoza, 50009 Zaragoza, Spain; hidalgo@unizar.es; 4GENUD (Growth, Exercise, Nutrition and Development) Research Group, University of Zaragoza, 50009 Zaragoza, Spain; borjamunizp@gmail.com

**Keywords:** elderly, resistance training, protein supplementation, muscle strength, physical performance

## Abstract

The aim of this study was to evaluate the effectiveness of the combination of resistance training (RT) and protein supplementation (PS), compared to RT alone or combined with a placebo (plS), in the improvement of muscle strength and physical performance. The search strategy in PubMed, Cochrane Library, and Web of Sciences databases found a total of 294 studies. Once inclusion and exclusion criteria were applied, 16 studies were included for the qualitative analysis. A total of 657 healthy elderly (>60 years) participants were analysed. Finally, 15 articles were included in the quantitative analysis with one being excluded due to issues with data availability. Upper-limb, lower-limb, and handgrip strength were the primary outcomes of the meta-analysis. The secondary outcomes, related to physical performance, were Short Physical Performance Battery (SPPB), gait speed, and the five-chair-rise test (5CRT). The main results of the meta-analysis show no statistical differences for upper-limb (SMD: 0.56, 95% CI: −0.09, 1.21, *p* = 0.09, I2 = 68%), lower-limb (SMD: 0.00, 95% CI: −0.18, 0.18, *p* = 1.0, I2 = 11%), and handgrip strength (SMD: 0.03, 95% CI: −0.26, 0.32, *p* = 0.84, I2 = 0%) between the RT + PS and the RT alone (or combined with plS). Moreover, no statistical differences were found relating to physical performance. In view of these results, protein supplementation combined with RT does not provide additional benefits compared to RT alone or with plS in healthy elderly adults.

## 1. Introduction

The ageing of the world’s population and the physical inactivity of older adults represent a major public health problem [1]. Lower mortality and increasing lifespan have led to a diversification and growth in chronic disease morbidity [2]. Such a trend includes an increased prevalence of aging-related mobility impairments, even with aging in the absence of disease [3]. Moreover, only 27–44% of older U.S. adults meet the World Health Organization’s (WHO) general recommendations for physical activity in adults [1,4]. In fact, individuals who did not meet this criteria have been found to have double the risk of future limitations in functional capacity [5].

Traditionally, sarcopenia has been defined as the muscle mass decrease related to aging [6]. However, the European Working Group on Sarcopenia in Older People (EWGSOP) recently stated that more attention should be given to a reduced muscle strength as the key characteristic to define and identify sarcopenia, with reduced muscle mass and physical performance taken as secondary factors [7]. Sarcopenia has an estimated prevalence of 10% in adults older than 60 years [3,8], rising to 50% in adults older than 80 years [3,8]. Studies show that elderly patients with lower muscle mass and strength have an increased probability of becoming dependent prematurely [9], longer and more frequent hospitalizations [10], and mortality [11], which in turn translates into higher healthcare costs [10].

Resistance training (RT) in the elderly population has been shown as the most useful tool for avoiding sarcopenia [3,12,13]. Moreover, RT alone or combined with other training methodologies has been demonstrated to be effective on the development of muscle mass, strength, and physical performance, as well as a decreased risk of fall in the physically frail elderly population [3,14].

Additionally, total protein intake seems to play an important role in sarcopenia [15,16]. However, protein supplementation (PS) alone has shown inconclusive results in its effectiveness to increase muscle mass, strength, and physical performance in sarcopenic population [17,18]. Further studies have shown positive effects of PS on muscle mass when combined with RT in older adults [19,20]. Given the importance of muscle strength and physical performance in the prevention and attenuation of sarcopenia, it is important to analyse whether combined RT and PS protocols are effective in improving these parameters [21].

The aim of this study is to evaluate if the combination of resistance training (RT) and protein supplementation (PS) is more effective than resistance training alone or combined with placebo (plS) in improving muscle strength and physical performance in healthy elderly adults.

## 2. Materials and Methods

### 2.1. Protocol and Registration

The study was performed following the Preferred Reporting Items for Systematic Reviews and Meta-Analyses (PRISMA) statement checklist [22]. The systematic review and meta-analysis protocol was registered in the Open Sciences Framework platform with the following DOI number: 10.17605/OSF.IO/CMU4R.

### 2.2. Information Sources and Search

The PICO (Population, Intervention, Comparison, Outcomes) strategy was taken into account in order to develop an accurate search strategy. The elected population was elderly people; the intervention studied was resistance training (RT) combined with protein supplementation (PS); the comparison chosen was resistance training alone or with placebo supplementation (plS); and the principal outcomes were physical performance (PP) and strength (ST). The search strategy was combined with randomized controlled trial (RCT) filters proposed by the Cochrane Collaboration [23]. The keywords used to develop the search are shown and classified by the PICO strategy on Table 1.

PubMed, Cochrane Library, and Web of Science were the databases used in this systematic review and meta-analysis. Moreover, manual searches and lists of references from additional studies were included, and other similar systematic reviews were checked in order to find potential studies that might meet the inclusion criteria. The final search was performed on June 5th, 2020. Table 2 shows the PubMed search strategy. The complete search strategies are available on the Appendix A.

### 2.3. Eligibility Criteria and Study Selection

Studies were included if they met the following criteria: (1) randomized controlled trial (RCT) study design, (2) adults aged >60, (3) healthy participants with or without sarcopenia condition, (4) intervention group with resistance training combined with protein supplementation, (5) comparison group with resistance training combined with placebo supplementation or no supplementation, (6) physical performance or strength as outcome, and (7) English language.

Studies were excluded if (1) the sample comprised hospitalized or post-surgery participants, (2) the sample comprised individuals with diabetes, cancer, cardiovascular disease, or other several acute/chronic conditions, (3) participants were using other forms of supplementation, such as vitamins, and (4) a resistance training duration of 8 weeks could not be completed.

Two authors (NL and LLL) independently screened titles, abstracts, and full text for potential inclusion. A third author (APB)was consulted in case of discrepancy. Inter-rater agreement was assessed by using Cohen’s Kappa index [24].

### 2.4. Data Collection Process

For each study included in this systematic review, the following data was extracted: (1) author’s last name, (2) year of publication, (3) sample size, (4) duration and frequency of the resistance training protocol, (5) type and dosage of protein supplementation, (6) control group protocol, (7) outcomes, and (8) main results.

### 2.5. Outcomes

The present systematic review and meta-analysis focused on two main outcomes: strength and physical performance. The assessment of the strength capacity was focused on those tests with a goal to determine the ability to generate high forces against large resistances. The assessment of physical performance included other speed or agility tests with greater coordinative demands.

### 2.6. Risk of Bias of Individual Studies

The Risk of Bias 2 tool (RoB 2) [25] from the Cochrane Collaboration and Physiotherapy Evidence Database (PEDro) scale [26] were used to assess the methodological quality and risk of bias of the randomized controlled studies included on this systematic review and meta-analysis.

The Risk of Bias 2 tool [25] from the Cochrane Collaboration is a domain-based evaluation that classifies seven domains from each randomized controlled trial into “low”, “unclear” or “high” risk of bias. The seven domains are based on publication bias (sequence generation and allocation sequence concealment), performance bias (blinding participants and personnel), detection bias (blinding outcome assessor), attrition bias (incomplete data), reporting bias (selecting outcome reporting), and other bias (e.g., sample size).

The PEDro scale [26] is an 11-item scale that relates the external validity (item 1), the internal validity (items 2–9), and the applicability or generalizability (items 10–11). One point is awarded if the criterion is clearly satisfied; thus, 11 points is the maximum score showing the highest methodological quality of a randomized controlled trial.

### 2.7. Statistical Analyses

The present meta-analysis was carried out using the RevMan Manager 5.3 software (the Cochrane Collaboration, London, UK, 2012). The sample size, means, and standard deviations (SD) for each variable were introduced in the software. If necessary, SD was calculated by standard error or confidence interval. All outcomes were continuous. If studies used different measuring tools, the chosen measure of effect size was Standard Mean Difference (SMD). On the other hand, if studies used the same measuring tool, Mean Difference (MD) was chosen as the effect size measure. An overall effect size with a 95% interval confidence (CI) was calculated. When studies did not report specific data (e.g., only graphs), an email was sent to the corresponding author asking for the missing data. If no response was received, the study was removed from the meta-analysis.

The SMD or MD of each outcome was calculated using a random-effects model (DerSimonian-Laird approach [27]). Heterogeneity across studies was evaluated by I^2^ statistics. Heterogeneity was classified as “small”, “moderate”, or “high” if I^2^ was <25%, 25–75%, and >75%, respectively, as Higgins et al. proposed [28]. The individual influence of each study on the overall result was analysed by removing each study once. Funnel plot visual interpretation was performed for outcomes with more than ten studies.

## 3. Results

### 3.1. Search Strategy

The search strategy found 294 studies (PubMed: 95; Cochrane Library: 105; Web of Science: 94). An additional study was included after performing the manual search. A total of 128 studies were excluded after checking and removing duplicates. A hundred and sixty-seven studies were initially considered to be included.

### 3.2. Study Selection

Firstly, the titles and abstracts of all included studies were screened, and 144 were excluded after this preliminary filter. Secondly, full text screening was carried out and 16 studies met the inclusion criteria for the qualitative analysis. One study was removed from the meta-analysis for not presenting the required data and after receiving no response from the corresponding author. Finally, 15 articles were included in the quantitative synthesis. Analysis of Cohen’s Kappa index showed a k = 0.83 categorized as “almost perfect” agreement [24]. PRISMA flow chart with detailed study selection is displayed in Figure 1.

### 3.3. Study Characteristics

The characteristics of the included studies are summarized in Table 3. The 16 studies included in this systematic review involved a total of 657 elderly individuals. Four studies included only women while five studies included only men. The other seven studies included both male and female participants. Sample sizes across studies ranged from 11 [29] to 141 [30].

Studies included in this systematic review involved participants from three continents. Two studies were from Asia, two studies were carried out in North America, three in South America, and nine studies in Europe.

Resistance training interventions had a duration of 12 weeks in most studies (*n* = 11) and the most common frequency was 3 times/week (*n* = 11). Whey protein supplementation was the most common across studies (*n* = 12). Fourteen studies assessed either lower-body or upper-body strength and 11 studies assessed function outcomes.

The control group of 14 studies received the same resistance training program plus a placebo supplementation. The remaining two studies had a control group receiving the same RT program without placebo supplementation. Full additional characteristics of the sample and the resistance training are available on Appendix A.

### 3.4. Risk of Bias Assessment

The methodological quality assessment by PEDro scale revealed a high quality across studies included in this systematic review. The average PEDro scale score was 8.5 points out of 11 (Table 4).

The RoB 2 tool summary and graph are shown in Figure 2 and Figure 3. Nine studies (56%) had at least four domains with “low risk”. Three studies (18%) had two or more domains as “high risk”.

### 3.5. Outcomes

The primary outcome assessed was strength (ST). It was often measured by using repetition maximum (RM) during leg extension for the lower extremity assessment, RM during chest press for the upper extremity assessment, and handgrip strength (HS). The secondary outcome was physical performance (PP). The main tests used to assess physical performance involved gait speed (GS), Short Physical Performance Battery (SPPB), and the five-chair-rise test (5CRT).

#### 3.5.1. Lower-Limb Strength (LLS)

Fourteen studies provided data about changes in lower-extremity strength. Thirteen of them assessed maximal quadriceps strength using a leg extension test (*n* = 12) or a leg press (*n* = 1). Among the 13 studies, nine followed the 1-RM method, two used an isometric dynamometer and two an isokinetic dynamometer. One study assessed the maximal voluntary contraction (MVC) of the plantar flexor muscles using surface electromyography. These studies involve a total sample size of 589 participants (298 in the protein supplementation + resistance training group and 291 in the resistance training group).

The overall standard mean difference was 0.00 with a 95% confidence interval of [−0.18, 0.18] and an overall effect of *p* = 1.0. The heterogeneity showed by the I^2^ statistic was low (I^2^ = 11%). Figure 4 shows the forest plot for lower-limb strength. Visual interpretation of the funnel plot reveals no evidence of publication bias for this outcome.

#### 3.5.2. Upper-Limb Strength (ULS)

Five studies provide data about changes in upper-extremity strength. The test used for this variable was a chest/bench press test. All studies assessed the maximal strength using the 1-RM method. A total of 66 participants were included in the intervention group (protein supplementation + resistance training) and 71 participants were included in the control group (only resistance training).

The intervention and control groups did not differ in upper-limb strength with an overall standard mean difference of 0.56 with a 95% confidence interval of [−0.09, 1.21] and an overall effect of *p* = 0.09. The I^2^ statistic revealed a moderate heterogeneity across studies (68%) (Figure 5). To investigate this factor, all studies were removed once from the analysis. When the Candow et al. [32] study was removed, the heterogeneity was 0%.

#### 3.5.3. Handgrip Strength (HS)

Only four studies provided sufficient data about handgrip strength involving 93 participants for the intervention group and 89 participants for the control group. The overall standard mean difference was 0.03 with a 95% confidence interval [−0.26, 0.32] and an overall effect of *p* = 0.84. The heterogeneity was I^2^ = 0% (Figure 6).

#### 3.5.4. Gait Speed (GS)

Five studies evaluated the physical performance by assessing participant’s gait speed. The intervention group and control group sample sizes were 92 and 90 participants, respectively.

Two studies evaluated gait speed in meters per second. However, three studies assessed it as the time taken to complete 10 m. Thus, the directionality of this data had to be opposed in the meta-analysis.

The overall standard mean difference was 0.11 with a 95% confidence interval [−0.18, 0.40] and an overall effect of *p* = 0.45. The heterogeneity across studies was 0% based on I^2^ statistics (Figure 7).

#### 3.5.5. Short Physical Performance Battery (SPPB)

Three studies were included in this outcome analysis involving 119 participants (62 intervention group and 57 control group). All studies used the same scale, so the Mean Difference was used as the effect size measure.

The protein supplementation plus resistance training group and resistance training group did not differ in terms of SPPB. The overall mean difference was 0.21 with a 95% confidence interval [−0.44, 0.85] and an overall effect of *p* = 0.53. Analysis by the I^2^ statistic revealed moderate heterogeneity (I^2^ = 50%). Removing the study from Amasene et al. [31] or the one from Holwerda et al. [33], the heterogeneity decreased to I^2^ = 0% (Figure 8).

#### 3.5.6. Five-Chair-Rise Test (5CRT)

Six studies measured physical performance by the five-chair-rise test. This analysis involved 108 participants in the intervention group and 104 participants in the control group. Trabal et al. [29] only provided data of the change from baseline, so higher values implied better results. However, five studies measured it as the necessary time to achieve five chair rises with higher values implying worse results. Therefore, the data directions of these studies needed to be opposed for the meta-analysis.

The overall standard mean difference was 0.16 with a 95% confidence interval [−0.12, 0.43] and overall effect of *p* = 0.26. The heterogeneity across the studies was I^2^ = 0 (Figure 9).

Finally, a subgroup analysis regarding sarcopenic or healthy elderly participants was performed in order to assess differences with overall sample results. However, all forest plots indicated no statistical subgroup differences (*p* < 0.05). Forest plot with detailed information about this analysis are available on the Appendix A.

Complete sensitivity analysis is also available in the Appendix A.

## 4. Discussion

The present article aims to summarize the effects of an RT intervention combined with PS compared to an RT intervention alone or combined with placebo supplementation on strength and physical performance in the healthy elderly population. This systematic review summarizes findings from a total of 16 studies and including a total of 657 participants.

To our knowledge, this is the first meta-analysis that compares RT plus PS with RT alone or plus placebo supplementation in a healthy elderly population. Notably, there are similar previous studies that compared different populations, such as a younger population [45,46] or elderly hospitalized people [47]; included a combination of other supplements, such as vitamin 3, omega-3, or a dietary intake modification [19,48,49,50,51]; or studied other outcomes related to body composition [52,53].

In view of the results, our meta-analysis shows no statistical differences between RT in combination with PS compared with RT alone or combined with placebo on upper- and lower-limb strength, handgrip strength, gait speed, and functional tests SPPB and 5CRT.

### 4.1. Muscle Strength

Handgrip strength is a simple and inexpensive way to assess muscle strength [54], and it has been established as a reliable tool to predict increased functional limitations, quality of life, and death [55,56]. Moreover, handgrip strength has a moderate correlation with the strength of other parts of the body [7]. In fact, chest and leg press exercises are also used for the assessment of muscle strength in the elderly population [57].

Our findings suggest that protein supplementation does not provide any greater benefit when compared to RT alone or combined with placebo in terms of muscle strength improvements in both lower and upper limbs and handgrip strength in healthy elderly adults. These results agree with previous meta-analyses conducted by Ten Haaf et al. [47], Finger et al. [49], and Morton et al. [51], but differ from those of Hou et al. [46] and Liao et al. [19].

We believe that the difference between our findings and these studies lies in that Hou et al. [46] and Liao et al. [19] included other supplements, such as vitamin D, and their study populations were aged >50 years [46] or included hospitalized people [19]. In that sense, it could be hypothesized that PS might only provide additional benefits to RT in frail people, who are characterised by greater losses of muscle mass which may limit muscle strength development.

It is important to note that heterogeneity across studies was low for lower-limb strength (I^2^ = 11%) and handgrip strength (I^2^ = 0%), but moderate for upper-limb strength (I^2^ = 68%). Heterogeneity for upper-limb strength could be explained by the study of Candow et al. [32]. The heterogeneity was 0% when it was removed. The study design proposed by Candow et al. [32] is the only one including a ST program shorter than 12 weeks, which is the duration recommended by the National Strength and Conditioning Association (NSCA) [3]. This study notably favors RT + PS with a standard mean difference [95% CI] of 2.41 [1.26, 3.55]. It is difficult to provide with consistent evidence-based explanations given the large heterogeneity of the results. It could be possible that the effects of the PS on upper-limb muscle strength appear in a shorter term (<12 weeks) and favor the combination RT + PS. However, as the other included studies have a longer duration (>12 weeks), the effects of the PS might be underestimated.

### 4.2. Physical Performance

Apart from strength, physical performance assesses the whole-body function related to locomotion and the individual’s health status [7,13]. Some of the most relevant functional tests are SPPB [58,59], gait speed [60,61,62], and 5CRT [63,64].

As suggested by a previous meta-analysis conducted by Ten Haaf et al. [47] and Hou et al. [46], our results also show that PS combined with RT is not more effective than RT alone or combined with placebo in developing physical performance improvement in healthy elderly adults. In contrast, when studying a frailer population (hospitalized, institutionalized, or community-dwelling elderly individuals with a high risk of sarcopenia or frailty and physical limitations), the results from Liao et al. [19] showed significant improvements in the RT + PS group. These findings suggest that frail people who have some physical impairments could benefit from protein supplementation, but not in the case of healthy people with no severe physical limitations.

### 4.3. Protein Supplementation

Current recommendations for daily protein intake range from 1.2 to 1.5 g/kg body weight/day for elderly active population [65,66]. These amounts can be achieved through diet or through protein supplementation. To date, there is no general recommendation on the appropriate protein supplementation dose, since it depends on the body composition of each individual, as well as on their physical fitness and health conditions. However, some studies indicate that the dose could be between 25 and 30 g of protein [67]. Following these data, in at least eight of the studies included in this review, the supplementation dose is less than 25 g, so its benefits may not be reflected.

In general, our findings support the idea that resistance training is one of the most effective strategies to prevent or delay frailty condition in an elderly population.

Finally, the absence of additional benefits of PS when combined to RT suggests that we should rethink if it is necessary to supplement with protein every elderly people without taking into account their fitness or their health status.

This meta-analysis has some limitations, mainly related to the heterogeneous characteristics (intensity, frequency, volume etc.) and duration of the resistance training programs, the different types and doses of protein intake, and the diversity of the methods used to assess muscle strength and physical performance. Moreover, a longer follow-up may have been interesting to analyse if differences are seen in the long term.

## 5. Conclusions

In view of our results, there is not sufficient evidence to support the use of protein supplementation when combined with resistance training in healthy elderly adults for improving muscle strength and physical performance. Protein supplementation combined with RT does not provide additional benefits compared to RT alone. Future primary studies are needed to analyse the different protocols of protein supplementation. Furthermore, studies with longer follow-up periods should be conducted in order to analyse possible differences over time

## Figures and Tables

**Figure 1 nutrients-12-02607-f001:**
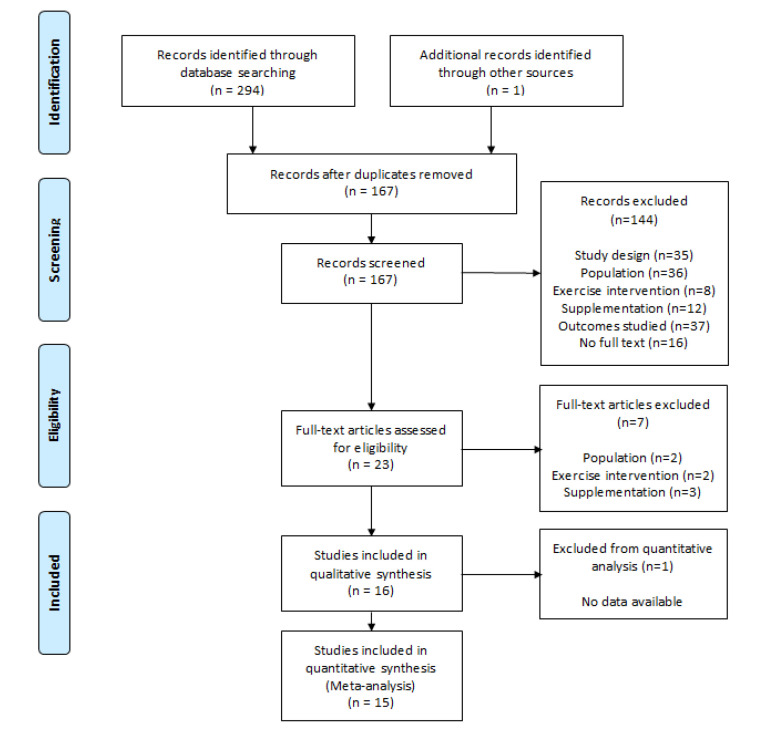
Search Preferred Reporting Items for Systematic Reviews and Meta-Analyses (PRISMA) flow diagram.

**Figure 2 nutrients-12-02607-f002:**
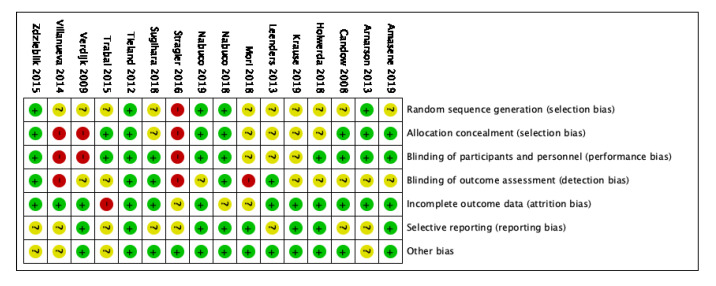
Risk of bias summary.

**Figure 3 nutrients-12-02607-f003:**
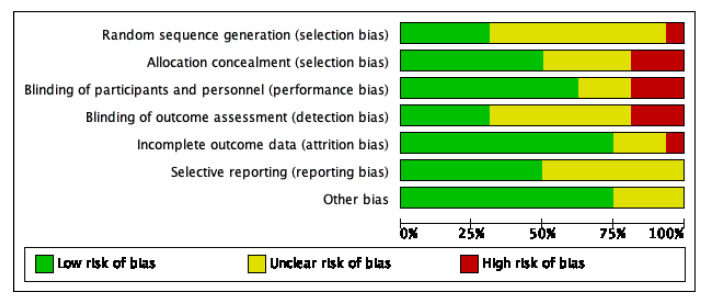
Risk of bias graph.

**Figure 4 nutrients-12-02607-f004:**
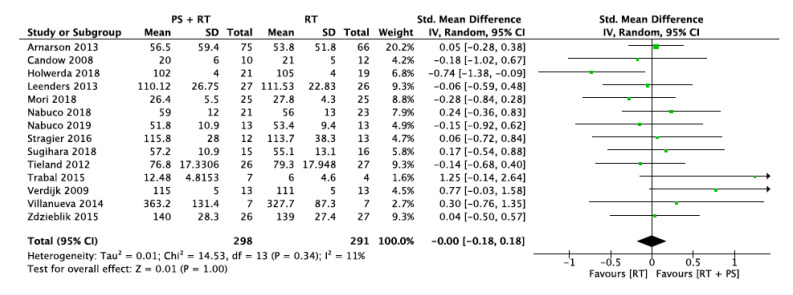
Impact of RT + PS on lower-limb strength.

**Figure 5 nutrients-12-02607-f005:**
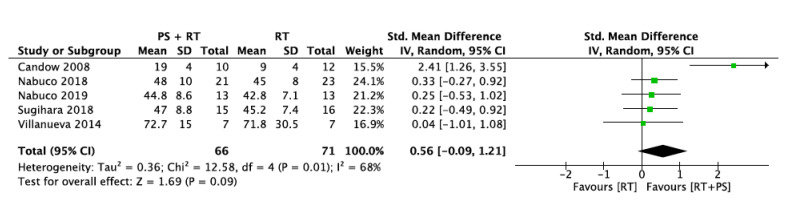
Impact of RT + PS on upper-limb strength.

**Figure 6 nutrients-12-02607-f006:**
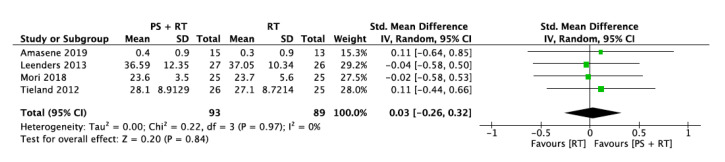
Impact of RT + PS on handgrip strength.

**Figure 7 nutrients-12-02607-f007:**
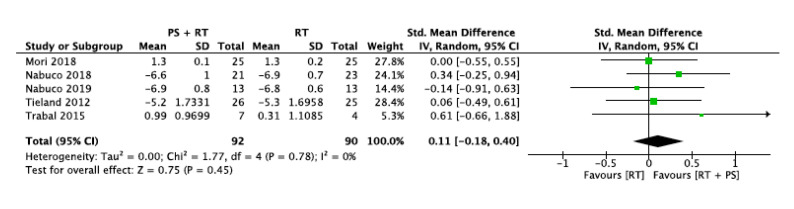
Impact of RT + PS on gait speed.

**Figure 8 nutrients-12-02607-f008:**
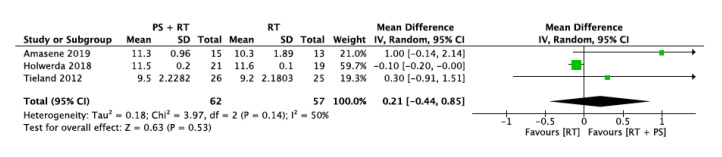
Impact of RT + PS on Short Physical Performance Battery (SPPB).

**Figure 9 nutrients-12-02607-f009:**
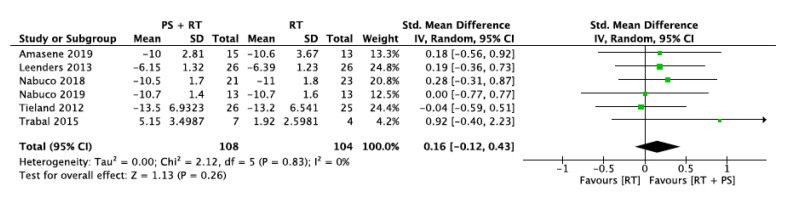
Impact of RT + PS on the five-chair-rise test (5CRT).

**Table 1 nutrients-12-02607-t001:** Keywords used for the search strategy.

Population	Intervention	Control	Outcomes
Aged	Resistance training	Protein supplementation	Physical fitness
Old people	Strength training	Supplemental protein	Functionality
Elderly			Performance
Aging			Fitness
Older people			Strength
Older adults			Resistance
Old adults			Endurance
Frail			Balance
Senior			Stability
Geriatric			Agility
			Mobility
			Gait
			Speed
			Locomotion
			Fall
			Falling
			Handgrip
			SPPB
			Tandem
			TUG
			Timed up and go
			Quality of life
			Min-mental
			Cognition

TUG: Timed Up and Go Test.

**Table 2 nutrients-12-02607-t002:** Search Strategy.

PubMed Search Strategy
(aged OR “old people” OR “older people” OR “older adults” OR “old adults” OR elderly OR senior OR geriatric OR frail) AND (resistance training OR strength training) AND (“protein supplementation” OR “supplemental protein”) AND (physical fitness OR functionality OR performance OR strength OR resistance OR endurance OR balance OR stability OR agility OR mobility OR gait OR speed OR locomotion OR fall OR handgrip OR SPPB OR tandem OR TUG OR “timed up and go” OR “quality of life”) AND ((randomized controlled trial [pt] OR controlled clinical trial [pt] OR randomized [TiAb]) OR placebo [TiAb] OR clinical trials as topic [mesh: noexp] OR randomly [TiAb]) OR trial [Ti]) NOT (animals [mh] NOT humans [mh]))

**Table 3 nutrients-12-02607-t003:** Characteristics of the studies included in the qualitative analysis.

Study	Population		RT Intervention	PS Intervention		CG	Outcomes	Main Results
	N (EG/CG)	Gender (M/F)	Duration × Frequency	Type	Amount (g/d or g/s)			
Amasene, 2019 [31]	28 (15/13)	14/14	12 w × 2 s/w	Whey (+Leucine enriched)	20 g (+3 g)/s	RT + plS	PP	ND PP
Arnarson, 2013 [30]	141 (76/66)	NR	12 w × 3 s/w	Whey	20 g/s	RT + plS	ST, PP	ND LLS ND PP
Candow, 2008 [32]	22 (10/12)	20/0	10 w × 3 s/w	Whey (+Creatin)	Protein: 0.3 g∙kg^−1^ Creatin: 0.1 g∙kg^−1^	RT + plS	ST	↑ULS ND LLS
Holwerda, 2018 [33]	40 (21/19)	40/0	12 w × 3 s/w	Whey (+Leucine enriched)	21 g/d	RT + plS	ST, PP	ND LLS ND SPPB
Krause, 2019 [34]	21 (11/10)	12/9	12 w × 3 s/w	Whey	0.165 g∙kg^−1^	RT + plS	PP	ND PP
Leenders, 2013 [35]	53 (27/26)	29/24	24 w × 3 s/w	Whey (+Casein)	3 g + 12 g/d	RT + plS	ST, PP	ND LLS ND 5CRT ND HS
Mori, 2018 [36]	50 (25/25)	0/50	24 w × 2 s/w	Whey	25 g/s	RT	ST, PP	ND LLS ND HS ND GS
Nabuco, 2018 [37]	44 (21/23)	0/44	12 w × 3 s/w	Whey	35 g/s	RT + plS	ST, PP	↑ST ↑PP
Nabuco, 2019 [38]	26 (13/13)	0/26	12 w × 3 s/w	Whey	35 g/s	RT + plS	ST, PP	ND ST ND PP
Stragier S, 2016 [39]	25 (12/13)	11/14	24 w × 2 s/w	Leucine	27.6 g/d	RT + plS	ST	ND LLS
Sugihara, 2018 [40]	31 (15/16)	0/31	12 w × 3 s/w	Whey	35 g/s	RT + plS	ST	↑ST
Tieland, 2012 [41]	53 (26/27)	NR	24 w × 2 s/w	Whey	15 g/d	RT + plS	ST, PP	ND LLS ND HS ND SPPB ND GS 5CRT
Trabal, 2015 [29]	11 (7/4)	NR	12 w × 4 s/w	Leucine	10 g/d	RT + plS	ST, PP	↑LLS ↑TUG ND 5CRT
Verdijk, 2009 [42]	26 (13/13)	26/0	12 w × 3 s/w	Casein	20 g/s	RT + plS	ST	ND LLS
Villanueva, 2014 [43]	14 (7/7)	14/0	12 w × 3 s/w	Whey (+Creatin)	35 g/d	RT	ST, PP	ND ULS ND LLS ND PP
Zdzieblik D, 2015 [44]	53 (26/27)	53/0	12 w × 3 s/w	Collagen peptides	15 g/d	RT + plS	ST	↑LLS

EG: Experimental Group; CG: Control Group; RT: Resistance Training; PS: Protein Supplementation; plS: Placebo Supplementation; ND: No significant differences; ↑: Significant increase for EG; w: week; d: day; s: session; ST: Strength; LLS: Lower-Limb Strength; ULS: Upper-Limb Strength; HS: Handgrip Strength; PP: Physical Performance; SPPB: Short Physical Performance Battery, GS: Gait Speed, 5CRT: Five-Chair-Rise Test; TUG: Timed Up and Go Test; NR: No Report.

**Table 4 nutrients-12-02607-t004:** PEDro scale.

Study	1	2	3	4	5	6	7	8	9	10	11	Total
Amasene, 2019 [31]	X	X	X	X	X				X	X	X	8
Arnarson, 2013 [30]	X	X	X	X	X	X		X		X	X	9
Candow, 2008 [32]	X	X	X	X	X	X		X		X	X	9
Holwerda, 2018 [33]	X	X	X	X	X	X		X		X	X	9
Krause, 2019 [34]	X	X	X	X	X			X	X	X	X	9
Leenders, 2013 [35]	X	X	X	X	X	X		X		X	X	9
Mori, 2018 [36]	X	X	X	X	X			X		X	X	8
Nabuco, 2018 [37]	X	X	X	X	X	X		X	X	X	X	10
Nabuco, 2019 [38]	X	X	X	X	X	X		X		X	X	9
Stragier S, 2016 [39]	X	X		X						X	X	5
Sugihara, 2018 [40]	X	X	X	X	X	X		X	X	X	X	10
Tieland, 2012 [41]	X	X	X	X	X	X		X	X	X	X	10
Trabal, 2015 [29]	X	X	X	X	X	X				X	X	8
Verdijk, 2009 [42]	X	X	X	X			X	X		X	X	8
Villanueva, 2014 [43]	X	X	X	X				X		X	X	7
Zdzieblik D, 2015 [44]	X	X	X	X	X		X	X		X	X	9
**Average**												8.5

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
