# Peer review of "Effectiveness of Protein Supplementation Combined with Resistance Training on Muscle Strength and Physical Performance in Elderly: A Systematic Review and Meta-Analysis"

_nutrients, 2020, doi:10.3390/nu12092607_

Round 1

Reviewer 1 Report

The manuscript by Labata-Lezaun et al., entitled “Effectiveness of protein supplementation combined with resistance training on muscle strength and physical performance in elderly…” is a systematic review of the literature including a meta-analysis to address the question whether protein supplementation has synergistic effects with resistance training on what is considered the most important parameter to assess skeletal muscle quality in sarcopenia, i.e. the muscle functional capacity. The authors ultimately considered 15 papers for the meta-analytical statistics, representing 657 elderly individuals. The conclusion is that protein supplementation does not add additional benefits to the resistance training for the elderly people.

I have mixed impressions on this manuscript, since it appears to be very interesting and well written, even though not completely novel in respect to recent, similar articles. Below, a more detailed explanation of what I mean.

In the last few years, several metanalyses on “resistance training” and “protein supplementation” have been published by good scientific journal, including Nutrients. The latter, in particular, has published 5 meta-analyses on this topic in the last 3 years (Nutrients. 2018 Feb 16;10(2):221; Nutrients. 2019 Sep 2;11(9):2047; Nutrients. 2019 Jun 25;11(6):1429; Nutrients. 2019 Sep 9;11(9):2163; Nutrients. 2018 Dec 4;10(12):1916)). A couple of these papers are actually cited. The authors write: “To our knowledge, this is the first meta-analysis that compares RT plus PS with RT alone or plus placebo supplementation in a healthy elderly population. Notably, there are similar previous studies that compared different populations, such as elderly hospitalized people; included a combination of other supplements, such as vitamin 3, omega-3, or a dietary intake modification; or studied other outcomes related to body composition or blood sample analysis”. However, based on my understanding of the literature, the conclusions of this manuscript are not novel. The authors should explain to the reviewers and the editors in more detail in what their manuscripts differs from the previously published articles and justify the need to further develop on the subject of resistance training and protein supplementation.

This being said, this work by Labata et al. is excellent on a formal point of view for a systematic review and metanalysis paper;  indeed, the manuscript fulfills the PRISMA guidelines for systematic reviews, which are similar to other Systematic Reviews and Meta-Analyses, i.e.: 1. Formulate the review question; 2. Define inclusion and exclusion criteria; 3. Develop search strategy and locate studies; 4. Select studies; 5. Extract data; 6. Assess study quality; 7. Analyze and interpret results (J Can Acad Child Adolesc Psychiatry. 2011 Feb; 20(1): 57–59).

Also, this manuscript addresses in a clear-cut way a very important issue, which paves the way to recommendations and clinical practices for the elderly people.

I have not specific comments nor minor remarks to do, since the manuscript per se is formally perfect.

Reviewer 2 Report

In the present manuscript, the authors investigated using a meta-analytic approach the effect of adding protein supplementation to resistance training intervention in healthy elderly. The primary outcome of the analysis was strength of the upper and lower limb, while the secondary one was physical performance (functional physical fitness). As the authors reported during their dissertation, previous research synthesis deal with efficacy of nutritional supplementation in ageing, however there was no consistency in the inclusion criteria (e.g. frail and not frail older adults). Similarly, different type of supplements other than protein were included, concealing the real treatment effect of the combination of protein and resistance training. For the previous reasons, I particularly appreciate the rationale of the study and its intent to shed light on a particular issue often overgeneralized.

To address their research question the authors used a solid methodology, facilitating the replication of the analysis. The presentation of the section is rigorous and understandable, however during reading some particular issue arose. I prefer to proceed in analytical manner to go straight to the point and be concise.

Major Concern

Why the author did not search EMBASE, SportDiscuss and PEDro? Access to the database should be limited for logistic reason but it should be stated clearly. That is because the aforementioned resources are often included in Resistance training research synthesis.

From my point of view the authors should discuss the reason why they decided not to use the thesaurus. The resources to carry out a bibliographic search has different tools (such as a specific thesaurus, the possibility to explode thesaurus structure, automatic term mapping,…) and, in my opinion, the exact motivation that bring authors to include the research term in the way reported need to be specified. Moreover, it should be specified the reason why the exact search was conducted in PubMed, Web of Science and Cochrane Library.

Table 1: Resistance training paradigm includes weightlifting (broader meaning) and calisthenics (body weight exercises). Why the author did not include in the research string any reference to the second component of resistance training? Resistance training was introduced as MeSH term in 2009 and literature paper did not use it frequently in the past. My opinion is that limiting the research string to “Resistance training” and “strength training” decreased the retrieval capacity of the research strategy. With these premises, I am asking to justify or address the mentioned points.

Data collection process: In my opinion, research synthesis and meta-analysis should describe the state of the art of the literature. Even if ES of the studies seems to be consistent and different treatment in term of resistance training seems to not influence the results, the reader of your work might be interested to know if you control for other variables respect to frequency and duration. My suggestion is to include when possible, at least in the appendix, a detailed table for each intervention in term of: i) baseline volume load, ii) final volume load, iii) distribution between multiple and single joint exercise, iv) mean rate of volume load  increase, v) proximity to failure/or reaching failure vi) volume split for different body part (if possible), vii) number of exercises. To better understand how to describe resistance training intervention, please see:

-Haff, G. G. (2010). Quantifying workloads in resistance training: a brief review. Strength Cond J, 10, 31-40.

- Scott, B. R., Duthie, G. M., Thornton, H. R., & Dascombe, B. J. (2016). Training monitoring for resistance exercise: theory and applications. Sports Medicine, 46(5), 687-698.

In the listed variables cannot be coded, it should be reported in the limitation section.

Data collection process: I need to express to the authors my concerns about the sample description. I appreciate the selection of specific subsample respect to other reviews, however it should be useful to provide a table that summarize some important information about the sample of the different studies. Particularly, if the original studies include the following data they should be included: i) eating habits of participant, ii) overall protein intake of participant, iii) percent of sarcopenic elderly in the sample. Moreover, it could be interesting to split the sample according to age class 65-75, 75-85, >=85.

Data collection process:Should be possible to separate sarcopenic to healthy elderly in the meta-analysis and to compare results with the overall sample?

Minor Concern:

To favour the results reading, I suggest moving table and figure after the specific paragraph. Table 3 after 3.3 study characteristics, table 4 figure 2 and figure 3 after 3.4 risk of bias assessment, figure 4 after 3.5.1 lower limb strength, figure 7 after 3.5.4 gait speed,

Line 22: PubMed and Web of Science are research engine not database.

Table 1: Which was the rational to include “Resistance” (quotation mark does not mean exact phrase search) in the “Outcomes” section of the research string?

Line 22 and 142: The capacity of retrieve 294 studies could highlight over specificity of the research string or missed results. My suggestion is to discuss this result respect to other similar review highlighting the reason why the number of studies retrieved is small. It should not be cause of the RCT filters alone.

Table 3: To better understand the type of resistance training protocol, I suggest to the authors to include a description of the interventions (weight-lifted, calistenichs, elastic bands, strength machines, …)

Table 4: Please, for the sake of clarity state explicitly the name of the domain of the PEDro scale not only in the method section but also in the table (intesar of 1-11).

Statistical analysis:How was handle the lack of information about allocation concealment in computing effect size? If post treatment mean was used, how was absence of information about absence of difference at baseline addressed?

Statistical analysis: the authors described in some results sessions the sensitivity analysis. Could be interesting adding the complete sensitivity analysis in a Appendix session.

New directions:

Did the author test the relationship between age and strength gain under protein+resistance training condition?

Round 2

Reviewer 2 Report

Thanks to reply adequately to my requests. I not have other requests.